# Set Norm and Equivariant Skip Connections: Putting the Deep in Deep Sets

## Abstract

Permutation invariant neural networks are a promising tool for predictive modeling of set data. We show, however, that existing architectures struggle to perform well when they are deep. In this work, we mathematically and empirically analyze normalization layers and residual connections in the context of deep permutation invariant neural networks. We develop set norm (SN), a normalization tailored for sets, and introduce the "clean path principle" for equivariant residual connections alongside a novel benefit of such connections, the reduction of information loss. Based on our analysis, we propose Deep Sets++ and Set Transformer++, deep models that reach comparable or better performance than their original counterparts on a diverse suite of tasks. We additionally introduce Flow-RBC, a new single-cell dataset and real-world application of permutation invariant prediction. We open-source our data and code here: link-omitted-for-anonymity.

## 1 Introduction

Many real-world tasks involve predictions on sets or multisets, from point cloud classification (Guo et al., 2020; Wu et al., 2015; Qi et al., 2017a) to the prediction of health outcomes from single-cell data (Regev et al., 2017; Lähnemann et al., 2020; Liu et al., 2021; Yuan et al., 2017). In the context of health prediction among other applications, there is the need for reliable methods that can be applied by application-based practitioners without the additional requirement of engineering models for every application task.

An important property of models applied to input sets is *permutation invariance*: for any permutation of the instances in the input set, the model prediction stays the same. Deep Sets (Zaheer et al., 2017) and Set Transformer (Lee et al., 2019) are two general-purpose deep permutation invariant models. Such models have been proven to be universal approximators of permutation invariant functions under the right conditions (Zaheer et al., 2017; Wagstaff et al., 2019). In practice, however, the architectures are tailored to specific tasks to achieve good performance.

In this work, we explore a more general approach to achieving good performance: making networks deeper, a strategy which has yielded benefit across many tasks (He et al., 2016b; Wang et al., 2019b). We first note that naively stacking more layers on both the Deep Sets and Set Transformer architectures hurts performance. Consider Figure 1. Deep Sets 50 layers (panel (a)) has significantly worse performance than Deep Sets 3 layers due to the problem of vanishing gradients. Residual connections are often used to combat vanishing gradients, but even the Set Transformer architecture which consists of residual connections has worse performance at a higher depth (panel (b)). Another strategy often used in combination with residual connections in deep architectures is the use of normalization layers, but the use of layer norm does not improve or deteriorates performance of both architectures at high depths, one with residual connections (Set Transformer) and one without (Deep Sets). In fact, the Set Transformer

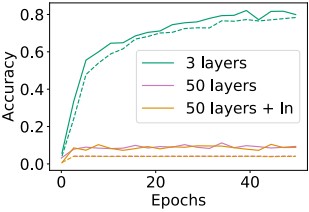

(a) Deep Sets

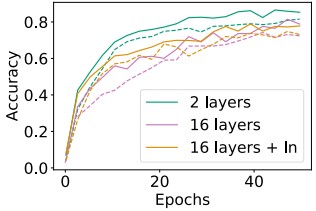

(b) Set Transformer

Figure 1: Deep versions of existing models perform worse.

paper (Lee et al., 2019) mentions layer norm as a component of their architecture but turns it off in actual implementation.

In this work, we tackle the challenge of designing deep permutation invariant networks with the aim of providing general purpose architectures.

- Based on real-world applications, we formalize **desiderata for normalization layers when inputs are sets** and systematically analyze options based on a unified framework. We introduce **set norm**, aimed at optimizing the desiderata we introduce. Set norm is a general, easy-to-implement normalization that can be applied on any architecture. We empirically demonstrate its benefit with ablation studies (Section 3, Section 5).

- We consider settings for residual connections which prevent vanishing gradients and maintain permutation equivariance. Through extensive experiments, we show that the best setting is that of **equivariant residual connections** (ERC) coupled the **"clean residual path" principle** (Section 4). To the best of our knowledge, we are the first to analyze this principle across architectural designs and to suggest a new benefit of certain residual connections: alleviation of information loss (Section 5).

- We develop **Deep Sets++ (DS++)** and **Set Transformer++ (ST++)**, leveraging set norm and ERC. We demonstrate that both models yield comparable or improved performance over their existing counterparts across a variety of tasks (Section 5). Among other results, these new architectures yield better accuracy on point cloud classification than the task-specific architectures proposed in the original Deep Sets and Set Transformer papers. The fact that set norm and equivariant residual connections yield improvements on two different architectures suggests that they may also be able to benefit other permutation invariant architectures in the future.

- Given growing interest in single-cell analysis (Regev et al., 2017) and the need for methods which can extract signal from single-cell data (Lähnemann et al., 2020), we introduce a **novel clinical single-cell dataset for prediction on sets** called **Flow-RBC**. The dataset consists of red blood cell (RBC) measurements and hematocrit levels (i.e. the fraction of blood volume occupied by RBCs) for 100,000+ patients. The size and presence of a prediction target (hematocrit) makes this dataset unique, even among single-cell datasets in established repositories like the Human Cell Atlas (Regev et al., 2017). Flow-RBC not only provides a new real-world benchmark for methodological innovation but also a potential source of information for biomedical science (Section 5).

## 2 BACKGROUND

Let $S$ be the number of samples in a set. A function $f : \mathcal{X}^S \to \mathcal{Y}$ is *permutation invariant* if any permutation $\pi$ of the input set results in the same output: $f(\pi x) = f(x)$. A function $\sigma : \mathcal{X}^S \to \mathcal{Y}^S$ *permutation equivariant* if, for any permutation $\pi$, the outputs are permuted accordingly: $\sigma(\pi \mathbf{x}) = \pi \sigma(\mathbf{x})$. A sum/max-decomposable function $f : \mathcal{X}^S \to \mathcal{Y}$ is one which can be expressed via an equivariant encoder $\sigma : \mathcal{X}^S \to \mathcal{Z}^S$ that maps each input sample in $\mathcal{X}$ to a latent space $\mathcal{Z}$, a sum or max aggregation over the sample dimension of the encoder output, and an unconstrained decoder $\rho : \mathcal{Z} \to \mathcal{Y}$:

$$f(\mathbf{x}) = \rho \left( \sum \sigma(\mathbf{x}) \right). \tag{1}$$

An architecture is a universal approximator of permutation invariant functions if and only if it is sum/max-decomposable with sufficient conditions on the latent space dimension (Wagstaff et al., 2019). Both the Deep Sets and Set Transformer architectures can be written in the form of Equation (1). Concretely, Deep Sets architecture consists of an equivariant encoder made up of feedforward layers, a sum or max aggregation, and a decoder also made up of feedforward layers (Zaheer et al., 2017). The Set Transformer encoder consists of multi-attention blocks called inducing point attention blocks (Lee et al., 2019), a variant of the original transformer block (Vaswani et al., 2017) modified to handle large set sizes. The aggregation is learned via attention, and the decoder consists of transformer blocks with self-attention.

Both architectures are permutation invariant since they consist of a permutation equivariant encoder and a permutation invariant aggregation over the outputs of the encoder. In order to maintain permutation invariance of the architectures upon modification, one needs to ensure that changes to the encoder preserve permutation equivariance, and changes to the aggregation maintain permutation

invariance. In the next two sections, we discuss how to incorporate normalization layers and residual connections while maintaining permutation invariance overall.

# 3   NORMALIZATION FOR SET DATA

Incorporating normalization layers into the permutation equivariant encoders requires careful consideration, as not all normalization layers are appropriate to use. We study normalization layers which consist of two operations, standardization and transformation. This setting captures most common normalizations (Ioffe & Szegedy, 2015; Ba et al., 2016; Ulyanov et al., 2016).

Let $\mathbf{a} \in \mathbb{R}^{B \times S \times D}$ be the activation before the normalization operation, where $B$ is the size of the batch, $S$ is the number of samples in a set (the tensor is zero-padded to the largest sample size), and $D$ is the feature dimension. First, the activations are standardized based on a setting $L$ which defines which dimensions utilize separate statistics. For instance, $L = \{B, S\}$ denotes that each set in a batch and each sample in a set gets its own mean and standard deviation for standardization, which are repeated over the $D$ dimension so that $\mu_L(\mathbf{a}), \sigma_L(\mathbf{a}) \in \mathbb{R}^{B \times S \times D}$ match the original activation tensor for elementwise subtraction and division. A standardization operation can be defined as:

$$\bar{\mathbf{a}}_L = \frac{\mathbf{a} - \mu_L(\mathbf{a})}{\sigma_L(\mathbf{a})}. \tag{2}$$

Next, the standardized activations are transformed through learned parameters which differ only over a setting of dimensions $M$. For instance, $M = \{D\}$ denotes that each feature is transformed by a different scale and bias, which are shared across the sets in the batch and samples in the sets. Let $\overrightarrow{\gamma}_M, \overrightarrow{\beta}_M \in \mathbb{R}^{B \times S \times D}$ denote the learned parameters and $\odot$ represent elementwise multiplication, any transformation operation can be defined as:

$$\hat{\mathbf{a}}_M = \bar{\mathbf{a}} \odot \overrightarrow{\gamma}_M + \overrightarrow{\beta}_M. \tag{3}$$

We can now summarize a wide array of normalization possibilities simply via choice of $L$ and $M$.

## 3.1   NORMALIZATION DESIDERATA

We formally introduce the desiderata for a general-purpose normalization layer for prediction on sets. In particular:

(i) The normalization must be permutation equivariant.

(ii) The computation of the statistics for standardization ($\mu_L(\mathbf{a}), \sigma_L(\mathbf{a})$) should not be affected by the presence of sets of varying sizes.

(iii) The normalization should not deteriorate performance due to removal of information that is useful for prediction.

Item (ii) is meant to address the concern that real-world set inputs are often not all the same size, e.g. not all point cloud objects are made up of the same number of points, and not all patients will have the same number of cells in a blood sample. If the presence of different set sizes forces a choice for an application-specific implementation of a normalization layer, then such a normalization is not general-purpose. Item (iii) refers to the fact that the standardization operation in a normalization layer incurs a loss of information, as activations which were once of varying means and variances across certain dimensions are now forced to be zero mean and unit variance. For real-valued sets in particular, this can greatly limit predictive performance, e.g. if the removed means and variances would help in the output prediction. As an example, the mean cell volume of a blood sample can be indicative of patient health, and a normalization layer that makes it impossible to distinguish between patients' mean cell volumes will restrict performance across a wide variety of health outcomes predictions.

## 3.2   THE ONLY SUITABLE TRANSFORMATION FOR SETS IS ON THE FEATURE DIMENSION.

Only normalization layers adopting a per feature transformation, i.e. $M = \{D\}$, meet the permutation equivariance desiderata (i), as per sample transformations break permutation equivariance and per

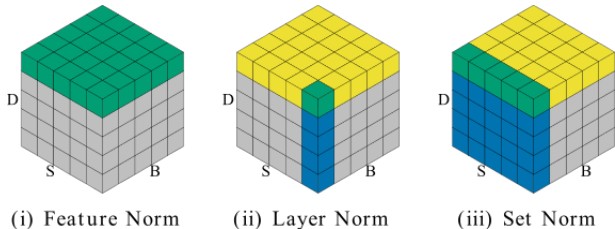

(i) Feature Norm     (ii) Layer Norm     (iii) Set Norm

Figure 2: We can represent normalization layers by the dimensions they perform standardization (blue) and the dimensions they perform transformation (yellow). Green indicates overlap of yellow and blue.

set transformations cause the prediction output to change when an element's position in the batch changes (see Appendix B for proofs).

**Proposition 1.** *Per feature transformation is the only setting which maintains permutation equivariance and prediction behavior which is agnostic to batch position.*

Thus, to systematically compare normalizations, we only need to consider the settings for $L$.

### 3.3 EXISTING NORMALIZATIONS DO NOT MEET ALL DESIDERATA.

We now consider possible settings for standardization: $\{\}, \{B\}, \{S\}, \{D\}, \{B, S\}, \{B, D\}, \{S, D\}$. We first note that $L = \{S\}$ and $L = \{S, D\}$ destroy permutation equivariance (see Appendix B, for proof). Of the remaining settings, we note that two already exist in the literature. The first, which we call *feature norm*, is used in the permutation invariant PointNet and PointNet++ architectures for point cloud classification (Qi et al., 2017a;b) and performs a per feature standardization followed by a per feature transformation, i.e. $L = \{D\}, M = \{D\}$ (see Figure 2 (i)). The second is layer norm, used in Set Transformer, which applies standardization individually to each set and each sample and transformation separately per feature, i.e. $L = \{B, S\}, M = \{D\}$ (see Figure 2(ii)).

Both these normalization layers have drawbacks. Feature norm calculates statistics over the batch and consequently may exhibit batch effects: sets in the batch are no longer independent, and performance can be heavily impacted by batch size (Hoffer et al., 2017). Additionally, like in batch norm, in feature norm sets are standardized differently at training and test time.[1] Moreover, in the context of sets of varying sizes, each possible implementation of feature norm (e.g. computing statistics over the batch, averaging statistics computed on each set separately) will lead to each set (or samples in the set) having a different impact in the computation of the statistics. Therefore, the use of feature norm requires careful consideration, especially in applications having as input real-valued sets with wide ranges of both set sizes and first moments. A similar property holds for normalizations which calculate statistics over the batch ($B \notin L$) such as $L = \{\}$.

Layer norm, on the other hand, removes information which may be particularly useful for set prediction tasks. By performing standardization on each sample in each set separately, layer norm maps two samples whose activations differ in only a scale and bias to the same output. The inability to distinguish such samples after layer norm can hurt prediction, as we see in Section 5.3.

**Proposition 2.** *Applying layer norm in its most common placement (after linear projection, before non-linearity) removes mean and variance information from each sample.*

The use of layer norm can present a representation issue similar to the oversmoothing problem in Graph Convolutional Networks (Li et al., 2018), where nodes (samples) get mapped to the same value as the number of layers in the network increase and neighborhoods of nodes become similar in representation.

---

[1]Feature norm is implemented as batch norm on a transposed tensor. Thus, test batches are standardized by a running mean and variance while training batches are standardized by their own batch statistics.

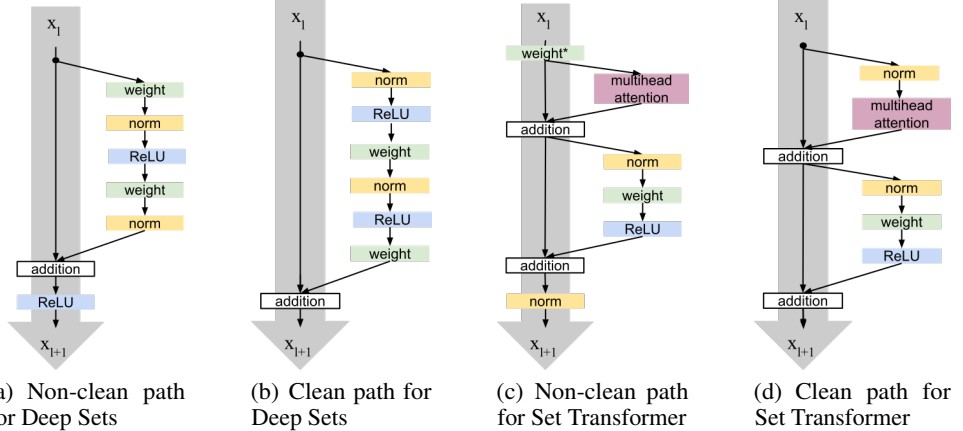

(a) Non-clean path for Deep Sets

(b) Clean path for Deep Sets

(c) Non-clean path for Set Transformer

(d) Clean path for Set Transformer

Figure 3: (Non-) clean path visual representation for Deep Sets and Set Transformer. In (c), weight* is also part of the attention computation.

### 3.4 SET NORM

We introduce a new normalization layer which exhibits a reasonable trade-off over the three desiderata. Our proposed normalization, set norm (SN), is a per set standardization, per feature transformation, i.e. $L = \{B\}, M = \{D\}$, see Figure 2(iii):

$$\text{SN}(a_{bsd}) = \frac{\mathbf{a}_b - \mu_b}{\sigma_b} \odot \gamma_d + \beta_d, \quad \mu_b = \frac{1}{S}\frac{1}{D}\sum_{s=1}^{S}\sum_{d=1}^{D} a_{bsd}, \quad \sigma_b^2 = \frac{1}{S}\frac{1}{D}\sum_{s=1}^{S}\sum_{d=1}^{D}(a_{bsd} - \mu_b)^2$$

SN is permutation equivariant (see Appendix B for proof) and, since is a per set standardization, it naturally handles sets of different sizes It also avoids batch effects that may arise when using feature norm while resulting in less information loss than layer norm. In fact, the per set standardization of set norm, i.e. $L = \{B\}$, removes the least information of any standardization which acts on each set separately (see Appendix B for proof). It is for this reason that we propose set norm over another per set standardization, e.g. per set per feature.

**Proposition 3.** *Among all standardizations which are applied separately per set, the per set standardization of set norm results in minimal loss of information.*

Experimental results (Section 5) confirm our hypothesis that set norm is generally preferable to feature norm and layer norm. However, set norm does remove some information, namely the global mean and variance of each set (as does any standardization with $B \in M$). In the next section, we discuss how we can alleviate this information loss through the use of residual connections.

## 4 RESIDUAL CONNECTIONS FOR SET DATA

Residual connections can be incorporated into an equivariant encoder in two ways: either adding each sample input to its corresponding output (equivariant residual connection) or adding an aggregated function over all the samples (aggregated residual connection).

An **equivariant residual connection** (ERC) is a residual connection that acts on every sample. Let $f$ be an equivariant function where $\mathcal{X} = \mathcal{Y} = \mathbb{R}^d$, i.e. $f : \mathbb{R}^{M \times d} \to \mathbb{R}^{M \times d}$. A function $g$ which adds single sample equivariant residual connections to any equivariant function $f$ is also permutation equivariant:

$$g(\pi\mathbf{x}) = f(\pi\mathbf{x}) + \pi\mathbf{x} = \pi f(\mathbf{x}) + \pi\mathbf{x} = \pi g(\mathbf{x}).$$

An **aggregated residual connection** (ARC) is a residual connection that sums a permutation-invariant aggregated function of the samples (e.g. sum, mean, max). A function $g$ which adds aggregated equivariant residual connections to any equivariant function $f$ is also permutation equivariant:

$$g(\pi \mathbf{x}) = f(\pi \mathbf{x}) + \text{pool}(\mathbf{x}_1, \ldots, \mathbf{x}_S) = \pi f(\mathbf{x}) + \text{pool}(\mathbf{x}_1, \ldots, \mathbf{x}_S) = \pi g(\mathbf{x}).$$

One major benefit of residual connections is that of addressing vanishing gradients. We show in Appendix F that ERCs and ARCs (mean and max) offer this benefit. While a sum aggregation does as well, we see empirically that such networks often explode in their outputs and therefore do not report results on it.

## 4.1 THE CLEAN PATH PRINCIPLE

There are many existing arrangements of residual pipelines (He et al., 2016b;a; Vaswani et al., 2017; 2018). We hypothesize that the best way to implement residual connections is to leave a clean path from input to output without other operations in between; we call this the *clean residual path* principle, where the residual path is the shortest path from input to output of a block (see Figure 3 for illustration). Our hypothesis aligns well with previous literature on non permutation invariant architectures showing how the presence of certain operations between skip connections could yield undesirable effects (He et al., 2016a; Wang et al., 2019a; Xiong et al., 2020) (see Appendix C for more details). Concretely, we propose the use of the pipeline in Figure 3(b) over the pipeline in Figure 3(a) for DeepSets, as well as the pipeline in Figure 3(d) over the one in Figure 3(c) for SetTransformer (He et al., 2016a;b; Klein et al., 2017; Vaswani et al., 2018; Xiong et al., 2020) for all possible implementation of residual connections that respect permutation invariance.

We additionally hypothesize a further benefit of the clean path principle, i.e. skip connections alleviate information loss due to normalization. While this is true in general, for applications involving images or text (the bulk of the DNN literature) information loss to normalization may be less important than specific data invariances. For prediction on real-valued sets like in distribution regression, on the other hand, this information is fundamental for good performance.

## 4.2 RESIDUALS CAN ALLEVIATE LOSS OF INFORMATION FROM NORMALIZATION

In specific tasks such as distribution regression, moments are often crucial to reach good performance. Applying normalization techniques that rely on standardization leads to the loss of information on the first two moments along some dimension. Clean path equivariant and aggregated residual connections will alleviate this issue as they help reintroduce such information.

Consider set norm placed before the nonlinearity in Deep Sets, e.g. $f(\mathbf{x}) = \text{relu}(\text{SN}(\mathbf{x}W))$ for a single layer. If two sets are scaled and translated versions of each other, i.e. $\mathbf{y} = a\mathbf{x} + b$, then without residual connections, the network with set norm will map both inputs to the same output, i.e. $f(\mathbf{x}) = f(\mathbf{y})$. On the other hand, Deep Sets with a residual connection (both aggregated or equivariant) will result in different outputs: $g(\mathbf{x}) = \mathbf{x} + f(\mathbf{x}) \neq g(\mathbf{y})$. Consider now layer norm in the same placement. If two samples in a set are mapped to the same output, an ERC will pass each sample forward, alleviating information loss, while ARCs will only pass the same aggregated function forward. Therefore, ERCs can alleviate information loss to a greater extent than ARCs can. This difference can have large implications for performance on some tasks (see Section 5.3). Moreover, ARCs could lead to similar representations across sample activations if the aggregated activations become larger in magnitude than the single sample ones they are added to. This resembles the concept of oversmoothing for graph neural networks (Li et al., 2018).

## 5 EXPERIMENTS

We run our experiment on tasks with diverse inputs (point cloud, continuous, images) and outputs (regression, classification). On these tasks we compare the original architectures with normalization and residual connection choices described in the previous sections. For all experiments, we fix the decoder and aggregation to their original versions, focusing on the effect of the encoder changes. Since the decoder is an unconstrained network, we expect general architectural principles non-specific

Table 1: Adding set norm and feature norm improves performance of Deep Sets (50 layers), while adding layer norm does not.

| | Hematocrit (MSE) | Point Cloud (CE) | MNIST Var (MSE) | Normal Var (MSE) |
|---|---|---|---|---|
| no norm | $25.8791 \pm 0.0014$ | $3.6090 \pm 0.0003$ | $5.5545 \pm 0.0014$ | $8.0228 \pm 0.0017$ |
| layer norm | $25.8748 \pm 0.0018$ | $3.6186 \pm 0.0001$ | $5.5651 \pm 0.0006$ | $7.9512 \pm 0.0026$ |
| feature norm | $25.8749 \pm 0.0018$ | $\mathbf{0.9940 \pm 0.0045}$ | $2.2629 \pm 0.2715$ | $\mathbf{0.3965 \pm 0.0059}$ |
| set norm | $25.8749 \pm 0.0018$ | $1.5423 \pm 0.0862$ | $\mathbf{0.2578 \pm 0.0031}$ | $5.4131 \pm 0.0081$ |

Table 2: The setting of Deep Sets which performs best overall uses set norm with a clean residual path (highlighted gray). First row is the original Deep Sets architecture.

| Path | Residual type | Norm | Hematocrit (MSE) | Point Cloud (CE) | Mnist Var (MSE) | Normal Var (MSE) |
|---|---|---|---|---|---|---|
| none | none | none | $25.8791 \pm 0.0014$ | $3.6090 \pm 0.0003$ | $5.5545 \pm 0.0014$ | $8.0228 \pm 0.0017$ |
| non-clean path | equivariant | layer norm | $19.6063 \pm 0.0137$ | $\mathbf{0.5982 \pm 0.0064}$ | $0.3528 \pm 0.0063$ | $0.1641 \pm 0.0386$ |
| | | feature norm | $19.9801 \pm 0.0862$ | $0.6541 \pm 0.0022$ | $\mathbf{0.3371 \pm 0.0059}$ | $0.4762 \pm 0.0832$ |
| | | set norm | $19.3146 \pm 0.0409$ | $\mathbf{0.6055 \pm 0.0007}$ | $\mathbf{0.3421 \pm 0.0022}$ | $0.2711 \pm 0.0813$ |
| clean path | equivariant | layer norm | $19.5236 \pm 0.0132$ | $0.6522 \pm 0.0072$ | $0.3529 \pm 0.0063$ | $0.1641 \pm 0.0386$ |
| | | feature norm | $19.3917 \pm 0.0685$ | $0.7148 \pm 0.0164$ | $\mathbf{0.3368 \pm 0.0049}$ | $0.1162 \pm 0.0177$ |
| | | set norm | $19.2118 \pm 0.0762$ | $0.7096 \pm 0.0049$ | $\mathbf{0.3441 \pm 0.0036}$ | $\mathbf{0.0320 \pm 0.0118}$ |
| clean path | mean | layer_norm | $21.0247 \pm 0.0503$ | $0.8656 \pm 0.0059$ | $1.1732 \pm 0.0259$ | $3.6164 \pm 0.1397$ |
| | | feature_norm | $\mathbf{18.9233 \pm 0.0638}$ | $0.9773 \pm 0.0288$ | $1.1296 \pm 0.0222$ | $1.9861 \pm 0.8488$ |
| | | set_norm | $19.3462 \pm 0.0260$ | $0.8585 \pm 0.0253$ | $1.2808 \pm 0.0101$ | $0.8287 \pm 0.1818$ |
| clean path | max | layer_norm | $21.9485 \pm 0.0181$ | $0.9637 \pm 0.0048$ | $1.3373 \pm 0.0124$ | $0.7133 \pm 0.0395$ |
| | | feature_norm | $21.0755 \pm 0.0440$ | $0.9092 \pm 0.0248$ | $1.4285 \pm 0.0174$ | $1.0224 \pm 0.0956$ |
| | | set_norm | $19.8171 \pm 0.0266$ | $0.8758 \pm 0.0196$ | $1.3798 \pm 0.0162$ | $0.9084 \pm 0.0910$ |

to sets to be applicable, and because of our focus is scaling architectures to large depths, we do not modify the aggregation, which itself is typically a single block.

## 5.1 ARCHITECTURES

To incorporate normalization layers and residual connections to Deep Sets and Set Transformer, we modify the equivariant encoder to consist of residual blocks (See Figure 3). For the clean path version of Deep Sets, we precede the first residual block with a linear layer and no bias and place a normalization-relu-weight sequence after the final residual block, following (He et al., 2016a). When we use equivariant clean path residuals with set norm, we called the architecture **Deep Sets++** or **Set Transformer++**. The clean path Set Transformer block architecture resembles that of the Pre-LN Transformer (Klein et al., 2017; Vaswani et al., 2018).

## 5.2 DATASETS & TASKS

We use four main datasets for ablations (Hematocrit, Point Cloud, Mnist Var and Normal Var) and one (CelebA) for validation of the models.

- **Hematocrit Regression from Blood Cell Cytometry Data (Hematocrit).** The prediction of hematocrit (the fraction of blood volume occupied be RBCs) from a set of individual RBC measurements (volume and hemoglobin) aims to answer an open biological and clinical question: are characteristics of individual cells predictive of their prevalence in the blood? On this task, increases in performance provide exciting scientific signal which can help lead to a deeper understanding of RBC production and regulation.[2] The dataset consists of 98240 train and 23104 test volume/hemoglobin distribution coupled with their hematocrit levels. We select the first visit for a given patient such that each patient only appears once in the dataset, and there is no patient overlap between train and test. We subsample for each distribution to 1,000 cells. See Appendix A for complete details about the dataset.

- **Point Cloud Classification (Point Cloud).** Following Zaheer et al. (2017); Lee et al. (2019), we use the ModelNet40 dataset (Wu et al., 2015) (9840 train and 2468 test clouds), randomly sample 1,000 points per set, and standardize each object to have mean zero and unit variance. We report ablation results as entropy loss to facilitate the readability of the tables, i.e. lower is better.

---

[2]As an analogy, imagine we learned for the first time that the distribution of individual leaf colors predicted the number of leaves on a tree. This insight would provide strong evidence for scientists to look for an underlying mechanism for this relationship, e.g. understanding how deciduous trees change in the fall.

Table 3: The setting of Set Transformer which performs best overall uses set norm with a clean residual path (highlighted gray). First row is the original Set Transformer.

| Path | Residual type | Norm | Hematocrit (MSE) | Point Cloud (CE) | MNIST Var (MSE) | Normal Var (MSE) | Improved tasks |
|---|---|---|---|---|---|---|---|
| non-clean path | equivariant | none | $18.7982 \pm 0.0086$ | $0.9217 \pm 0.0119$ | $6.2663 \pm 0.3307$ | $\mathbf{0.0015 \pm 0.0001}$ | - |
| non-clean path | equivariant | layer norm | $19.0904 \pm 0.1003$ | $0.9219 \pm 0.0052$ | $2.0663 \pm 1.0039$ | $0.0597 \pm 0.0157$ | 1/4 |
| | | feature norm | $19.4968 \pm 0.1442$ | $0.8251 \pm 0.0025$ | $\mathbf{0.4043 \pm 0.0078}$ | $0.0884 \pm 0.0092$ | 2/4 |
| | | set norm | $19.0521 \pm 0.0288$ | $1.9167 \pm 0.4880$ | $\mathbf{0.4064 \pm 0.0147}$ | $0.0627 \pm 0.0217$ | 1/4 |
| clean path | equivariant | layer norm | $18.8857 \pm 0.0583$ | $0.6656 \pm 0.0148$ | $0.6383 \pm 0.0020$ | $0.0053 \pm 0.0015$ | 2/4 |
| | | feature norm | $19.1967 \pm 0.0330$ | $\mathbf{0.6188 \pm 0.0141}$ | $0.7946 \pm 0.0065$ | $0.0224 \pm 0.0061$ | 2/4 |
| | | set norm | $\mathbf{18.6883 \pm 0.0238}$ | $\mathbf{0.6280 \pm 0.0098}$ | $0.7921 \pm 0.0006$ | $0.0068 \pm 0.0008$ | **3/4** |
| clean path | mean | layer_norm | $19.8620 \pm 0.0963$ | $0.8745 \pm 0.0079$ | $1.2562 \pm 0.0611$ | $0.1123 \pm 0.0158$ | 2/4 |
| | | feature_norm | $19.3714 \pm 0.1143$ | $0.8657 \pm 0.0106$ | $1.8169 \pm 0.0052$ | $0.1407 \pm 0.0398$ | 2/4 |
| | | set_norm | $19.6945 \pm 0.1067$ | $0.8111 \pm 0.0453$ | $1.6273 \pm 0.0335$ | $0.0512 \pm 0.0179$ | 2/4 |

Table 4: Deep Sets++ and Set Transformer++ outperform their Deep Sets and Set Transformer counterparts at 50 and 16 layers respectively.

| | Hematocrit (MSE) | Point Cloud (CE) | MNIST Var (MSE) | Normal Var (MSE) | CelebA (accuracy) |
|---|---|---|---|---|---|
| DeepSets (3 layers) | $\mathbf{19.1257 \pm 0.0361}$ | $0.7357 \pm 0.0119$ | $0.4520 \pm 0.0111$ | $\mathbf{0.0417 \pm 0.0074}$ | $0.3808 \pm 0.0016$ |
| Deep Sets (50 layers) | $25.8791 \pm 0.0014$ | $3.6090 \pm 0.0003$ | $5.5545 \pm 0.0014$ | $8.0228 \pm 0.0017$ | $0.1005 \pm 0.0000$ |
| DeepSets ++ (3 layers) | $19.5882 \pm 0.0555$ | $\mathbf{0.6703 \pm 0.0093}$ | $0.5895 \pm 0.0114$ | $0.0707 \pm 0.0326$ | $\mathbf{0.5730 \pm 0.0016}$ |
| Deep Sets++ (50 layers) | $\mathbf{19.2118 \pm 0.0762}$ | $0.7096 \pm 0.0049$ | $\mathbf{0.3441 \pm 0.0036}$ | $0.0320 \pm 0.0118$ | $\mathbf{0.5763 \pm 0.0134}$ |
| Set Transformer (2 layers) | $18.8750 \pm 0.0058$ | $0.7487 \pm 0.0381$ | $\mathbf{0.6151 \pm 0.0072}$ | $\mathbf{0.0016 \pm 0.0005}$ | $0.1292 \pm 0.0012$ |
| Set Transformer (16 layers) | $\mathbf{18.7436 \pm 0.0148}$ | $0.9217 \pm 0.0119$ | $6.2663 \pm 0.3307$ | $\mathbf{0.0015 \pm 0.0001}$ | $0.4570 \pm 0.0540$ |
| Set Transformer++ (2 layers) | $18.9223 \pm 0.0273$ | $\mathbf{0.6366 \pm 0.0004}$ | $1.1525 \pm 0.0158$ | $0.0050 \pm 0.0008$ | $\mathbf{0.6533 \pm 0.0012}$ |
| Set Transformer++ (16 layers) | $\mathbf{18.7258 \pm 0.0342}$ | $\mathbf{0.6280 \pm 0.0098}$ | $0.7921 \pm 0.0006$ | $0.0068 \pm 0.0008$ | $\mathbf{0.6587 \pm 0.0016}$ |

- **Variance Prediction, Image Data (MNIST Var).** We implement empirical variance regression on MNIST digits as a proxy for real-world tasks with sets of images, e.g. prediction on blood smear or histopathology slides. We sample 10 images uniformly from the training set and use the empirical variance of the digits as a label. Test set and training set images are non-overlapping. Training set size is 50,000 sets, and test set size is 1,000 sets. We represent each image as 1D vector.

- **Empirical Variance Prediction, Real Data (Normal Var).** Each set is a collection of samples (sample size 1000) from a univariate normal distribution. Means are drawn uniformly in [-10, 10], and variances are drawn uniformly in [0, 10]. The target for each set is the empirical variance of the samples (regression task) in the set. Training set size is 10,000 sets, and test set size is 1,000 sets.

- **Supervised set anomaly detection (CelebA).** Following Zaheer et al. (2017), we construct 18,000 sets of 10 CelebA images (each $64 \times 64$) where one does not belong. For each set, we randomly select two attributes out of the 40 total and select 9 images which share both attributes and one which does not exhibit either. Train and test do not contain the same individuals.

Results are reported in Mean Squared Error (MSE) on the test set for the regression experiments and in cross entropy loss (CE) for point cloud classification, averaged over three seeds. We fix all hyperparameters, including epochs, and use the model at the end of training for evaluation. We notice no signs of overfitting from the loss curves. For further details, see Appendix D.

## 5.3 MAIN RESULTS

**Set norm has better performances than feature norm and layer norm.** We compare how feature norm, layer norm, and set norm affect performance of a deep Deep Sets model (50 encoder layers) in Table 1. In general, set norm and feature norm both improve results compared to no normalization while layer norm often does not. We hypothesize that the problem is primarily information loss and explore in Appendix E.4 why layer norm may be suitable for categorical text data and not for real-valued sets. A similar phenomenon seems to exist for set norm on Normal Var, but we show next that we can ameliorate the effect with residual connections. Feature norm greatly improves performance on Point Cloud, which we speculate is due to specific invariances that hold for clouds of points. We provide an experiment in Appendix E with additional evidence to support this hypothesis. While feature norm performs well in Table 1 in particular, set norm tends to perform better overall once other considerations such as residual connections are taken into account (Table 2, Table 3). Additionally, set norm does not require additional considerations under sets of varying sizes. This makes it the best choice for a general purpose architecture.

**Equivariant residual connections following the clean path principle have best performance overall.** Experiments in Table 2 and Table 3 confirm that clean path ERCs pipelines generally yield the best performance across set tasks, with ERCs performing better than ARCs both for Deep Sets (Table 2) and Set Transformer (Table 3). The effect of clean path ERCs is especially evident on Deep Sets with set norm on Normal Var: the clean path residual (3e-2) especially improves performance over the non-clean path residual (1e-1) and no residual (5.4), a testament to the ability of a clean path residual pipeline to make up for the loss of global scale information by the set norm operation. Clean path ARCs perform worse than both non-clean and clean path ERCs. We hypothesize that this may be due to a problem similar to oversmoothing. Note that table 3 does not have max aggregation results as at least one seed failed for all experiments due to problems in training.

**Deep Sets++ and Set Transformer++ outperform existing architectures.** Table 4 shows that deep DS++ and ST++ generally yield better or comparable performance to deep and shallow Deep Sets and Set Transformer. We note that depth plays a role in the improved performance, as shallow DS++ and ST++ do not improve results in general. When deep, Deep Sets++ consistently improves over Deep Sets, sometimes with orders of magnitude improvement in test loss. Set Tranformer++ improves performance on Point Cloud and MNIST Var and reaches comparable performance on Hematocrit and Normal Var. In Appendix E, we show on an official point cloud benchmark repository (Goyal et al., 2021a) that DS++ and ST++ without any modifications outperform versions of Deep Sets and Set Transformer tailored for point cloud classification (87% vs 86% accuracy). On the unseen set anomaly task (CelebA), both DS++ and ST++ perform significantly better than their counterparts. On Hematocrit, results surpass a model that uses the clinical baseline as well as prediction from moments computed on the distribution. The best performing model is ST++ with 18.73 MSE, a 28% error reduction over the clinical baseline (25.85 MSE).

## 6 RELATED WORK

There exist several permutation invariant neural architectures in the literature, including Deep Sets (Zaheer et al., 2017) and Set Transformer (Lee et al., 2019) for general-purpose sets and PointNet (Qi et al., 2017a) and PointNet++ (Qi et al., 2017b) specialized for point cloud data. Previous efforts to design residual connections (He et al., 2016b; Veit et al., 2016; Yao et al., 2020) or normalization layers (Ioffe & Szegedy, 2015; Ba et al., 2016; Santurkar et al., 2018; Ghorbani et al., 2019; Luo et al., 2019; Xiong et al., 2020; Cai et al., 2021) have often been motivated by particular applications; our work is the first to design residual connections and normalization layers for prediction on sets in general. Efforts to build deep neural architectures exist in computer vision (He et al., 2016b), natural language (Wang et al., 2019b), and graphs (Li et al., 2019; Chen et al., 2020; Zhao & Akoglu, 2019). While graph convolutional networks also encode permutation equivariance, in this paper we address distributional real-valued data as well as sets of high-dimensional inputs where no external information about graph structure is available.

## 7 CONCLUSION

We introduce set norm (SN) and equivariant residual connections (ERCs) and illustrate the benefits of both in comparison to other choices of normalization layers and residual pipelines. Using set norm and equivariant residual connections, we developed Deep Sets++ and Set Transformer++, deep permutation invariant architectures which improve upon the current state-of-the-art models. Such architectures are general-purpose architectures and the first permutation invariant architectures of their depth that show good performance on a variety of tasks without task-specific architectural changes. Interestingly, between Deep Sets++ and Set Transformer++, there is not a best choice in architecture across tasks, so the choice for which to use may often come down to computational resources and time. Set Transformer++ with 16 layers has more expensive computational and memory requirements than Deep Sets++ with 50 layers, so Deep Sets++ may be preferred under limited resources. Lastly, we introduced a new open-source dataset, Flow-RBC, which provides a real-world application of permutation invariant prediction in clinical science. With data from over 100,000 patients, including an external clinical measurement (i.e. hematocrit), Flow-RBC is uniquely suited among single-cell datasets to be a benchmark for deep predictive modeling on sets. We believe our new models and dataset have the potential to motivate future work and applications of prediction on sets.

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
