# OpenReview forum: "Set Norm and Equivariant Skip Connections: Putting the Deep in Deep Sets"
_ICLR.cc/2022/Conference — ICLR 2022 Submitted_

### Official Review · Reviewer_Dj6B · 2021-10-27

**Correctness:** 2
**Technical Novelty And Significance:** 2
**Empirical Novelty And Significance:** 2
**Recommendation:** 5
**Confidence:** 4

**Main Review:**

I appreciate the effort to tackle the challenge of designing deep networks. Indeed,  I feel that it is an important research question.
Overall the paper is well written and easy to follow. The formulation of the method is presented in a general form which allows the reader to easily understand the core details.
I appreciate the effort to introduce an additional benchmark, which will be publicly available to the community.

However, I do have some concerns regarding the current version of the paper, detailed next.

The authors state that “existing architectures struggle to perform well when they are deep”.  However, I am not convinced that the claim is True as existing architectures do incorporate normalization layers. Moreover, I think the authors should consider refining the original question to  two separated questions:

1) Do normalization and skip connections help networks in learning ? regardless of the choice of how many layers are there.
2) Is a deep architecture a good inductive bias for some tasks involving sets?

Figure 1 does not answer the first question as it does not apply normalization+skip layers to standard architectures.  It also does not fully answer the second question as it does not evaluate on different depths of networks. To my understanding, the main implication from Figure 1 is that with 50 layers and without normalization the learning performs poorly, which is not novel.

Moreover, existing architectures already incorporate normalization layers. For example, in the pointnet [1] paper, which is a similar architecture to deepsets,  they use normalization layers (feature normalization). Do they improve/stable/disimprove with respect to network depth? Is there a difference between set normalization to feature normalization? Moreover, the results in table 2 do not show a clear advantage for the set norm over the feature norm.

*Relation to graphs*

The current paper focuses on sets. Note that there is extensive literature on the more general settings of deep permutation invariant networks, which is the case of graph neural networks. For example, what is the relation between the proposed set norm and the one suggested in [2]? I would expect a more detailed discussion about this challenge in graphs versus sets.

*Residual connections*

Incorporating residual connections in set networks is not novel. It seems that some prior works have used it, as the adaptation of it to set networks seems natural. For example, see implementation details in the supplementary material of [3]. Few alternatives to clean path residual connections are discussed. Do the authors know of such works that follow these alternatives? What are the advantages of these alternatives? Are there any tasks which they improve? If the answer is no, then it is not clear why are they discussed.

*Requirements for a design of a normalization layer*

The authors identify some requirements for normalization layers of permutation equivariant networks.  What does “work well” in the context of varying set sizes mean?  What does essential information mean? It feels like these requirements are too general and should be refined. Moreover, it seems that some discussion relating to test time versus train time equivariance is missing. Is it true that the normalization choices that the authors show that fail (such as L = {B} )  in satisfying equivariance is only at training time? As in test time, some constant inferred from the batch statistics is used. If that is the case, then I expect to see a discussion on why satisfying the equivariant property in training time is important.

*Results*

For classification tasks, it would be more informative to report accuracy. For regression tasks, the units of the measurement should be reported and explained as well.

[1] Qi, Charles R., et al. Pointnet: Deep learning on point sets for 3d classification and segmentation.

[2] Zhao, Lingxiao, and Leman Akoglu. Pairnorm: Tackling oversmoothing in gnns.

[3]  Mescheder, Lars, et al. Occupancy networks: Learning 3d reconstruction in function space.


**Summary Of The Paper:**

The paper tackles the problem of designing deep networks that operate on sets. The challenge is addressed by normalization layers and skip connections. The authors formulate some design choices and advocate the choice of set normalization and clean residual connections. The layers are incorporated with two different choices of architectures and are evaluated on several sets learning benchmarks. In addition, a new benchmark is introduced.


**Summary Of The Review:**

* Well written and easy to follow
* The list of requirements identified by the authors from normalization equivariant layers needs a major refinement.
* Lack of novelty and the benefit of the set norm is unclear
* Missing discussion on the relation to graph neural networks
* The experiments do not test properly the effect of depth on learning

---

> ### Author Response · Authors · 2021-11-19
> **Author Response to Dj6B (Part 1)**
>
> Thank you for the many helpful questions and comments! Addressing each point-by-point (2 parts):
> 1. *[Not convinced about making the “existing architectures struggle to perform at high depth” claim given existing architectures incorporate normalization layers]*  Thank you for your comment. We have modified the introduction and Figure 1 to tighten this claim. In particular, we show in a modified Figure 1 that Set Transformer exhibits poor performance at high depths even with its normalization layer (layer norm). In fact, the authors of the Set Transformer paper seem to turn off normalization in their open-source implementation on Github. The fact that a deep Set Transformer, with or without layer norm, performs worse than its shallow counterpart motivates the need to study normalization layers and residual connections more carefully for permutation invariant architectures.
> 2. *[Reframing the problem as two questions: 1) do normalization and skip connections help learning? 2) is deep a good inductive bias?]*
> Thank you for the suggestion. We added experiments to answer both questions and have modified the text accordingly:
>     - *[Do normalization and skip connections help networks in learning regardless of depth]* We added results on shallow models which suggest that the benefit of normalization layers and residual connections primarily pertains to deeper models.
>     - *[Is a deep architecture a good inductive bias]* We notice in general that a deeper architecture leads to better results (see Table 4 in the paper), suggesting that deep architectures are indeed a good inductive bias even for simple tasks.
> 3. *[Effect of normalization on depth in PointNet]* We agree that this is an interesting question. However, the original PointNet and PointNet++ architectures consist of multiple types of blocks, so answering this question comprehensively requires considering many possible ways to make these architectures deeper (e.g. pick a segment of the architecture to scale up, scale each block separately, stack multiple sequences of blocks). As these architectures are designed for point cloud classification in particular rather than applicable to sets in general, we believe it is outside the scope of this paper to study them in depth.
> 6. *[Difference between set norm and feature norm]* Based on results overall (e.g. Tables 2 and 3 in the updated paper), set norm seems to generally perform better than feature norm. Moreover, as mentioned in the section on normalization (section 3), feature norm exhibits a strange and possibly harmful property under sets of varying sizes: in the computation of the batch statistics, it will either weight sets or their individual samples differently depending on set sizes and implementation. As set norm is a per set normalization, it does not exhibit this behavior. However, as we note in the paper, the benefit of one over the other may sometimes be dictated by task-specific characteristics (further discussion can be found in the first paragraph of the main results).
> 7. *[Relation to GCN]* Thank you for the great suggestion. We have drawn parallels to work in graph neural networks in the updated submission. In particular, we note a connection between our “loss of information” idea and the concept of oversmoothing in graph neural networks.
> 8. *[Relation between set norm and pair norm]* In our language of L and M, pair norm separates out the mean computation (L_m) from the variance computation (L_v). Whereas set norm is L = {B}, M={D}, pair norm is L_m = {B, D}, L_v = {B}. Pair norm does not have a learned transformation step. Instead, they set a single scalar hyperparameter C, shared across the network layers and all graphs, which enforces that the overall variance across the nodes at each layer is C. This constraint, in combination with the fact that the graph convolution operation makes connected nodes more similar in representation, results in connected nodes becoming closer in representation while disconnected nodes get further apart as the input proceeds through the network. This effect is irrelevant for sets since there is no concept of which nodes are connected vs. disconnected.
> 9. *[Alternatives to clean path residual connections]* Thank you for the question. The primary reason why we discuss the clean path principle is because the set transformer architecture does not use it, and we find that this simple modification improves test performance across tasks (see Table 4). We agree that the idea of the equivariant residual connections (ERCs) seems natural; however, it is not the only option that maintains permutation equivariance, and in our updated submission, we compare with other alternatives (e.g. aggregated residual connections) to further discuss the benefit of ERCs over other choices. 1/2

---

> ### Author Response · Authors · 2021-11-19
> **Author Response to Dj6B (Part 2)**
>
> 8. *[Requirements for a design of a normalization layer should be refined]* Thank you for your suggestion, we refined the phrasing for the desiderata. We update the sets of varying sizes requirement to emphasize that a general-purpose normalization layer should handle sets of varying sizes without additional design considerations (e.g. how the implementation affects the relative importance of samples or sets in the standardization computation). We have also reworded the “essential information” clause to be more precise: we wish to remove as little information as possible in case it is important for prediction.
> 9. *[Permutation invariance at train and test time]* We have interpreted the reviewer’s question to be whether certain normalization layers break equivariance only at train and not test time. In general, this is not the case. Permutation equivariance is broken at both train time and test time for the L = {S} setting, which we have shown in the appendix. Permutation equivariance is preserved in the L = {B} case (e.g. set norm) at both train and test time. Apologies in advance if we have misinterpreted your question; we are happy to reply if you are willing to ask it again!
> 10. *[Accuracy, units of measurement]* Thank you for the suggestion. We have added accuracy where relevant. To make the multi-experiment tables easy to read, we chose to use the loss (e.g. cross entropy, mean squared error) so that all the numbers can be consistently interpreted (e.g. lower is better everywhere).
>
> We also summarize overall paper changes and discuss novelty and significance in our [general response](https://openreview.net/forum?id=MDT30TEtaVY&noteId=k7r0mjV6pXq). 2/2

---

### Official Review · Reviewer_ifHM · 2021-10-30

**Correctness:** 4
**Technical Novelty And Significance:** 3
**Empirical Novelty And Significance:** 2
**Recommendation:** 5
**Confidence:** 4

**Main Review:**

Strengths
+ I enjoyed the discussion on design decisions around normalization layers and agree that set norm is better than what was used in the Set Transformer paper because it discards less information. Furthermore, this intuition is confirmed by experiments in tables 2~4, which show that set norm improves performance.
+ The discussion about having a clean residual path is interesting, and the paper draws a compelling parallel to design choices in ResNet models.

Weaknesses
- None of the results have error bounds, so it's hard to tell whether the differences are statistically significant. This problem is exacerbated because loss scales vary widely between tasks (10^1 Hematocrit, 10^-2~-4 Normal Var). Looking at the learning curves in Figure 1, the vanilla and ++ versions of some networks seem to have similar performance on a few tasks.
- The paper increases the number of layers for both architectures, and this is never addressed in the experiments. For example, what is the performance of Set Transformer++ with 2 layers, compared to the vanilla version?

Minor comments
- Figure 1: the caption says that the figures have (grey, orange, blue) lines, but the colors seem closer to (pink, orange, green). This was slightly confusing.
- The rightmost two columns of Table 4 were hard to understand at first. Maybe it's better to have two rows with the performance numbers for "Original" and "No Norm"?
- I think Table 5 should be moved to the main paper, to show that permutation-invariant networks compare favorably to traditional approaches in this real-world setting.

**Summary Of The Paper:**

This paper presents improved versions of two standard premutation-invariant networks: Deep Sets++ and Set Transformer++. They make two critical design decisions: set normalization and clean residual paths. They also introduce Flow-RBC, a new benchmark for permutation-invariant prediction. Finally, the paper evaluates several variants on several benchmarks, including Flow-RBC.

**Summary Of The Review:**

This paper proposes interesting design choices for permutation-invariant networks, but I cannot tell whether the empirical results are statistically significant. Additionally, the paper does not show the effect of their components on the original shallow networks. I'd be happy to increase my score if these issues are addressed.

---

> ### Author Response · Authors · 2021-11-19
> **Author Response to ifHM**
>
> Thank you for your review and suggestions. We have summarized paper changes/additions and addressed novelty and significance in the [overall response](https://openreview.net/forum?id=MDT30TEtaVY&noteId=k7r0mjV6pXq). Addressing your other points:
> 1. *[Shallow architectures results]* Thank you for the suggestion, we added results on the shallow architectures in the Table 4 which show that the benefit of adding set norm and equivariant residual connection is present only at high depths.
> 2. *[Figures and tables]* Thank you for the suggestion, we updated Figure 1 and Table 4. While we could not fit in Table 5 from the Appendix due to space, we mention the results of Table 5 in the main text when we discuss hematocrit results.

---

### Official Review · Reviewer_eZF1 · 2021-11-03

**Correctness:** 2
**Technical Novelty And Significance:** 3
**Empirical Novelty And Significance:** 2
**Recommendation:** 3
**Confidence:** 4

**Main Review:**

The main weaknesses are: lack of statistical analysis of the experimental evaluation and limited novelty of the pre-norm changes.

Lack of statistical analysis of the experimental evaluation: the authors use relatively small datasets which seem to exhibit high variance. Single run results presented by the authors are inconclusive and cannot be used to support claimed contributions. The authors could use bootstrapping or train multiple models with different random seed and report mean and std for all baseline and candidate models.

Limited novelty of the pre-norm changes: pre-norm ResNet and pre-norm Transformer are very common, the proposed addition of pre-norm block to DeepSet and SetTransformer does not seem significant.

Another weakness is the usage of synthetic datasets. Instead of synthetic datasets for variance prediction the authors could evaluate their changes on established benchmarks for predictions on sets like object detection (e.g. COCO dataset) and trajectory prediction (e.g. Argoverse dataset). In object detection models which operate on sets of input features and predict sets have been gaining popularity since DETR [1], whereas in trajectory prediction VectorNet [2] is an example of a regression model with set inputs. Both are very similar to SetTransformer, employ layer normalization, residual connections and could potentially benefit from Set Norm and pre-norm Transformer.

Notes to authors:
- each dataset description in section 6 needs a clear definition of task solved (classification/regression), size of train/val splits, loss functions used, and metrics used to evaluate
- figure 1 lacks x and y labels for each subfigure
- figure 3 would benefit from short subcaption for each subfigure
- all tables lack column metric definitions. It is not clear what the presented numbers mean, which are classification errors (or accuracy?) and which are mean squared errors
- I could not understand Appendix A, the description of Flow-RBC dataset. An explanation in layman's terms would be very helpful. More formally, a description of what input set elements and targets are in the dataset is required.

[1] Carion et al. End-to-end object detection with transformers
[2] Gao et al. Vectornet: Encoding hd maps and agent dynamics from vectorized representation

**Summary Of The Paper:**

The paper aims to improve training of deep neural networks applied to sets. The authors investigate permutation equivariant normalizations and residual connections in DeepSet and SetTransformer models, and propose a modification of layer norm which calculates standardization statistics over both features and all set elements. They also propose a modification to DeepSet and SetTransformer models inspired from pre-norm ResNet and pre-norm Transformer. The modifications are experimentally tested on a pointcloud classification dataset, synthetic variance regression, and a new dataset for set prediction contributed by the authors.

**Summary Of The Review:**

The contributions of the draft are empirical and the experimental evaluation lacks an analysis of statistical significance, so the contributions cannot be evaluated conclusively. At least one contribution has questionable novelty.

---

> ### Author Response · Authors · 2021-11-19
> **Author Response to eZF1**
>
> Thank you for your concerns. We address them below:
> 1. *[Limited novelty of pre-norm changes]*
> While our proposed clean path residual Set Transformer closely resembles the pre-norm Transformer, our analysis is new. First, the benefit of these residual connections has not been previously documented on the architectures and applications we consider (i.e. Set Transformer, which utilizes inducing point attention blocks that differ from self-attention, and prediction tasks on sets of various forms). Additionally, we posit a novel benefit of these residual connections (alleviating information loss) which may help explain the superior performance of clean path equivariant residual connections over other connections, e.g. non-clean path, aggregated residual connections (see general response and paper for more).
> 2. *[use of synthetic datasets, benchmarking on COCO and Argoverse datasets]* While object detection and trajectory prediction are important real-world applications, we believe they fall outside the scope of predictions on inputs which are inherently collections of examples. Object detection involves the prediction of sets as output, but it does not in general involve a collection of examples as input. In fact, although DETR passes in sets of features to the transformer portion of the model, it incorporates positional embeddings to explicitly break permutation equivariance and maintain spatial ordering between the input features. Moreover, the authors show through ablations that this positional information is critical for good performance, distinguishing this task from those with sets as input. We also do not consider graph neural networks and tasks on relational graph data since these models and tasks explicitly use external information about the relationships between objects. In contrast, we only consider tasks where this information is absent (e.g. only a collection of inputs in a set), or models which do not explicitly use this information (e.g. do not exploit an adjacency matrix of relationships).  We agree with the reviewer that benchmarking on real-world datasets is important, which is what motivated us to open source Flow-RBC.
> 3. *[figures, tables, experiments]* Thank you for your suggestions, we applied all the changes in the updated version of the document.
> 4. *[Flow-RBC description]* We modified the description of the dataset to be better suited to any general reader.
> We have summarized additional paper changes in our [general response](https://openreview.net/forum?id=MDT30TEtaVY&noteId=k7r0mjV6pXq), where we also further discuss novelty and significance.

---

### Official Review · Reviewer_Qf1Y · 2021-11-06

**Correctness:** 4
**Technical Novelty And Significance:** 3
**Empirical Novelty And Significance:** 3
**Recommendation:** 6
**Confidence:** 4

**Main Review:**

## Strengths

**S1) Overall well-written and clear**

The motivation for developing set-norm is clear, and the comparison to feature norm and layer norm is also clear. The motivation for the
"clean path" principle for residual connections is presented well and intuitive.

**S2) Interesting new dataset**

Admittedly, as I have no medical background, I am not a good adjudicator on the "value" of the new Flow-RBC dataset. However, taking the authors at their word, it seems to be much larger than existing similar datasets and demonstrates a useful real-world test case for set-based neural networks, beyond more commonly seen point cloud data and synthetic toy datasets (e.g., sum of MNIST digits).

## Weaknesses

**W1) What is special about an equivariant residual connection? And what is the relationship between the "clean path principle" and equivariance?**

It is unclear to me how an "equivariant residual connection" differs from a regular residual connection. To me, it seems like the discussion of the "clean path principle" is independent of equivariance and applies generally to all residual connections. If so, this needs to be made clearer in the paper.

**W2) Could use more empirical comparisons**

In particular, both of the original Deep Sets and Set Transformers paper do set anomaly detection on CelebA. It would be helpful to see a comparison of Deep Sets++ and Set Transformers++ on this task

**W3) Error bars on experimental results**

Were the experiments run once, or many times? Could you provide error bars for the experiments?

## Clarity Issues

- Figure 2: should the "F" axis labels be "D"?

- Please add units to all of the tables (or at least the table captions). I know that the units (MSE) are written in the main text, but they can be hard to find when looking just at the tables.

**Summary Of The Paper:**

The paper has four main contributions:

1. introduces the _set norm_ normalization layer in neural network models for set data, as opposed to feature normalization (aka. batch-norm) and layer normalization layers;
2. provides intuition behind a "cleaner" implementation of residual connections;
3. implements set norm and the cleaner residual connections in modified versions of existing Deep Sets and Set Transformers models;
4. introduces a new dataset called Flow-RBC with over 100,000 examples. Each "example" consists of a $(X,y)$ pair, where $X=\\{x_1, \dotsc, x_{1000}\\}$ is a set of measurements on 1000 red-blood cells from a patient, and $y$ is a hematocrit level label. Each $x_i$ consists of both volume and hemoglobin mass measurements. This dataset is significantly larger than similar existing datasets.

**Summary Of The Review:**

The paper is well-written and provides solid empirical justification for specific choices of normalization (set norm) and residual connection design ("clean path principle") for neural networks that operate on sets. While "set norm" is somewhat novel, I am unsure about whether the "clean path principle" applies specifically to these set-based neural networks, or whether there is something specific about that design for equivariant networks. Overall, a solid empirical paper, but I hesitate to say that it is truly "novel." That said, I am open to increasing my score based on authors' replies.

---

> ### Author Response · Authors · 2021-11-19
> **Author Response to Qf1Y**
>
> Thank you for your questions and suggestions! We have summarized paper changes/additions and addressed novelty and significance in the [overall response](https://openreview.net/forum?id=MDT30TEtaVY&noteId=k7r0mjV6pXq). Regarding other points:
> 1. *[equivariant residual connection vs. residual connection]* An equivariant residual connection (ERC) generalizes plain residual connections to equivariant functions, passing forward for each sample the input to be added to the output. In our update, we contrast ERCs to other connections which also maintain equivariance overall, namely aggregated residual connections.
> 2. *[clean path principle applies generally to all residual connections, independent of equivariance]*
> We agree that the clean path principle is not unique to ERCs and have made this clearer in the text. We emphasize the clean path principle since residual connections implemented in existing permutation invariant architectures (i.e. Set Transformer) do not abide by this principle, and we can improve performance by modifying the architecture based on the principle.
> 3. *[Adding CelebA set anomaly detection]*
> We have added experiments on CelebA set anomaly detection. DS++ and ST++ significantly improve over their original counterparts.
> 4. *[Figure 2 axis, units in table]* Thank you for pointing these out. We have addressed both.

---

### Author Response · Authors · 2021-11-19
**General Response**

Thank you to all the reviewers for your detailed comments and helping us improve our paper! Based on reviewer feedback, we made the following changes:

1. We rework the introduction to make the problem and our work’s significance more prominent. In particular, we note that careful design of residual connections and normalization is required to build successful deep architectures, as even the Set Transformer architecture which includes both often performs worse at deeper depths.
2. We clarify the significance of the Flow-RBC dataset with discussion that is more accessible to a general audience.
3. We highlight the systematic analysis which went into our normalization proposals and tighten the desiderata we pose.
4. We show that equivariant residual connections are not just a natural option for residual connections but in fact work better than other forms of residual connections which also preserve permutation equivariance and deal with vanishing gradients. To make this point, we add empirical comparisons with aggregated residual connections, which add the same aggregated function of the inputs to the output of the block.
5. We benchmark on a new task, CelebA set anomaly detection, and show that our proposed changes yield benefit.
6. We add experiments in the shallow network setting where the benefits of our proposed methods are minimal, suggesting that our methods primarily help with the task of making deeper models better.
7. We report mean and standard error over three seeds across all our results.
8. We draw comparisons with literature on graph neural networks. In particular, we note the connections between our discussion on loss of information and the discussion around oversmoothing in graph neural networks.

We also rephrased the text to accommodate the many additions and stay within nine pages. We have responded to all individual comments directly in their respective threads. Below is our general response to concerns about novelty and significance.

First, we would like to highlight that the novelty of our work goes beyond our proposed normalization layer (set norm) and residual connection setting (clean path equivariant residual connections). In particular,
1. We formalize general desiderata for normalization layers on tasks with sets as input.
2. We draw parallels between residual connections for disparate architectures, unified under the “clean path” principle.
3. We present a new potential benefit of clean path residual connections, alleviating loss of information, which is especially relevant to real-valued set inputs.

Next, we wish to emphasize that the primary significance of the paper lies in its general applicability, i.e. the methods to achieve good performance on multiple architectures and a variety of tasks. Unlike in the Set Transformer and Deep Sets papers, which perform task-specific engineering to achieve good performance across a diversity of tasks, we use the same architecture for all tasks. We believe that this generality  is fundamental for applying permutation invariant models at a large scale across tasks and can have wide impact especially given its ease of use among application-based practitioners.

We appreciate that reviewers found our work “well written” (Qf1Y, Dj6B) with “interesting design choices” (ifHM) while addressing “an important research question” (Dj6B). We hope the above helps clarify what we see to be the novelty and significance of this work.

---

### Decision · Program_Chairs · 2022-01-20

**Decision:**

Reject

**Comment:**

The submission focuses on a set norm normalization layer for neural network models, which stands in contrast to a batch norm.  The majority of the reviewers felt that this submission is not suitable for publication at ICLR in its current form.  These concerns remained after the post-rebuttal discussion.  Quoting from the reviewer discussion, the following points remained as significant concerns:


1. lack of novelty. normalization layers have been used previously in other sets architectures. The systematic approach for normalization in the current paper is nice but I am not sure how valuable it is. There exists already extensive literature on normalization layers for graphs (a generalization of sets).
2. lack of motivation for deep networks. The main claim in the paper (see the introduction and figure 1) is that 50 layers should perform better and since it does not seem to be the case in figure 1, it requires studying normalization layers. I am not sure it is a well-established claim. What are the assumptions leading to the conclusion that deep architectures should perform better on the task considered in figure 1? I am also concerned that normalization layers have a major effect on improving "not so deep" networks with 3-10 layers and not only the extreme 50 layers case, making the comparison in the paper between only 3 and 50 layers not enough for telling the full story. Thus evaluation on more depths is required.

On the balance, the paper does not meet the threshold for acceptance in this round of peer review.